# Insights and protocols for discrimination of sugarcane clones by dissimilarity measures on RGB and NIR data

**Luiz Alexandre Peternelli** *, **Andréa Carla Bastos Andrade**

Department of Statistics, Federal University of Viçosa, Viçosa, MG, Brazil

* peternelli@ufv.br

## Abstract

In sugarcane breeding, dense experiments have been considered in the initial phase (T1), such as the Simplified System (SS) of genotype selection. In this method, the seedlings of each family are transplanted directly from the seed box to the field, forming a kind of carpet. Despite the practical aspect of the method, selection problems are common, as stalks from the same individual within the family are subject to being taken to later evaluation stages, to the detriment of stalks from different individuals. To facilitate the discrimination of stalks of the same family in SS, we evaluated using RGB images (red:green:blue) and NIR (near infrared) spectra. We applied Euclidean distance (D) and Mahalanobis distance ($D^2$) dissimilarity measures to the image and spectral data to distinguish stalks with different genotypes. RGB and NIR data were taken from type +1 leaf samples collected from two experimental blocks, totaling 31 evaluated families. The analyzes were carried out in two stages. In the first stage, we sought to evaluate the classification capacity using RGB images and NIR spectra, using D as a measure of dissimilarity. In the second step, we developed and validated a protocol using RGB images to classify clones, with $D^2$ as a dissimilarity measure. Preliminary results, with distance D, allowed to discriminate clones based on the distance of the evaluated attributes and their combinations. In addition, with the analyzes using the D distance, it was identified that only the use of the R attribute (red band) would give satisfactory results for the second stage, which was the proposed analysis protocol, applying the $D^2$ distance. The $D^2$ statistic and associated p-value confirmed the protocol's usefulness in discriminating stalks in SS, especially stalks from the same families.

## Introduction

The selection of superior genotypes within a sugarcane population is a long-term task, with at least ten years to generate results. On average, one new variety is obtained for every 250 thousand seedlings evaluated in the first stage of the breeding program [1]. Within the genetic breeding programs of this crop, the conventional selection system is the primary way to obtain improved varieties [2]. However, in the conventional method, evaluating a high number of genotypes in the initial stages of breeding is impractical as financial and physical limitations, such as labor and experimental area, may occur [3].

**Data Availability Statement:** We have uploaded the dataset on a public repository and provided the

DOIs necessary to access these data. The DOI number is 10.6084/m9.figshare.23638932.

**Funding:** LAP; (CNPQ) - process 309662/2019-2; www.cnpq.br; The funders had no role in study design, data collection and analysis, decision to publish, or preparation of the manuscript.

A new genotype selection methodology, called Simplified System (SS) [4], has been evaluated to minimize the difficulties of the conventional system [5]. The method consists of planting in the field densely, or otherwise, all seedlings are transplanted directly from the sowing box to the field, forming a kind of carpet. The genotypes are selected a few months after planting in the field, selecting the individuals with the best vigor [6].

This new selection methodology has been introduced in some of Brazil's sugarcane genetic breeding programs. However, due to the conditions of density in the field caused by this selection system, it may be challenging to select the genotypes precisely without having identical duplicates when selecting vigorous individuals close to each other and from the same plant. Thus, it is necessary to develop procedures that help differentiate these genotypes in SS.

A possible procedure for the correct identification of different stalks is to carry out the phenotypic evaluation of samples in a non-destructive, fast, accurate, and precise way using digital cameras, sensors, and automatic mechanical devices [7] or using near-infrared (NIR) instruments [8]. In the literature, many studies have evaluated the application of NIR and RGB (Red:Green:Blue) spectroscopy to infer better fruit maturation [9,10] study changes in color and appearance in processed foods [11], evaluate viral infection in plants [12], as well as to differentiate related species [13]. Through these techniques, it is also possible to discriminate genotypes of the same species, as reported in the studies by [14–17].

In general, these studies use multivariate analysis techniques for comparison purposes. These analyzes assist in differentiating samples to identify those biologically relevant spectral characteristics [18]. Among the multivariate methods used, such as principal component analysis, discriminant analysis, and clustering, emphasis is placed on the use of dissimilarity measures, such as Euclidean distance and Mahalanobis distance, among others [19]. These statistical approaches applied to NIR and RGB image information can be very valuable in using SS in the initial phase of sugarcane breeding programs, maximizing gains in selection processes.

Given the above, the present work aims to: i) evaluate the potential use of NIR instruments and RGB cameras to distinguish sugarcane stalks as to their origin in the experimental plot; ii) establish a protocol for data collection and a quick and non-destructive analysis that infers about the origin of the stalk samples within densely populated families, which is the case of the SS, that is, if the samples of vigorous stalks that are close within the plot are obtained from genetically different individuals, that is, different clones.

## Material and methods

### Plant material

Sugarcane clones were evaluated in an experiment conducted in the Simplified System (SS) of the Sugarcane Genetic Breeding Program of the Federal University of Viçosa (PMGCA-UFV), reported in [5]. In this study, each block had plots containing a particular genotype selected from a given SS family. We collected samples from two experimental blocks for the present study. In block 1, images from 14 clones were obtained, while in block 2, images from 24 clones were obtained. There were 31 families, of which 7 had representatives in both blocks. From each plot containing a single clone, three leaves of the "+1" type (the first leaf wholly detached from the plant sheath) [20,21] were collected, considering those two experimental blocks (Fig 1).

Table 1 represents the classification form for the set of samples obtained for analysis.

### Obtaining RGB (Red:Green:Blue) and NIR (near-infrared) data

Leaf samples were collected in the field and identified according to the family and block of origin. Leaf samples were collected in the same week to ensure their same health and vigor

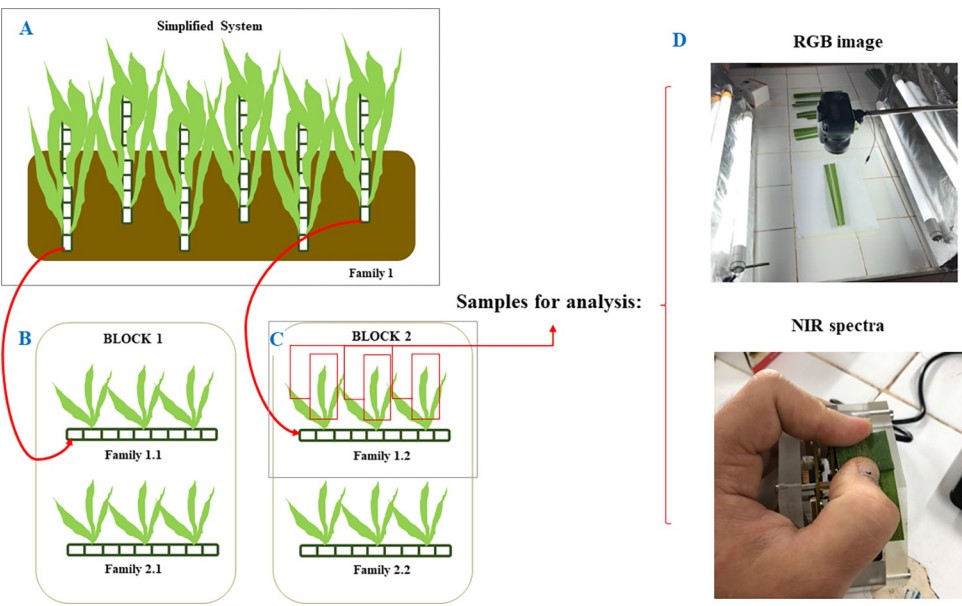

**Fig 1. Scheme of the experimental procedure for the origin of samples and data collection.** (A): Dense planting of seedlings from a given family i (i = 1 to F, the number of families evaluated) in the Simplified System; (B) the stalks of the most vigorous individuals in each family were selected and planted in new plots; in this case each, h block contains an individual selected from each family i; (C) three +1 leaves were randomly collected from each plot; (D) from each leaf, RGB and NIR data were obtained.

conditions. These samples were taken to the laboratory to obtain images and NIR data. First, RGB images were obtained from the middle third of the leaves. For this, a Fujifilm S4000 digital camera with 14 MegaPixels and a 30x Superwide lens was used 20 cm above the leaf. Leaves were previously positioned on a bright plane and between two light reflectors built to homogenize the lighting of the photos obtained (Fig 1D). Since the camera-to-object distance conditions were constant, we used the digital camera automatic option. Image segmentation was made automatically from the R script.

Additionally, using the DLP®NIRscan™ Nano EVMO spectrometer (Texas Instruments Inc., Dallas, Texas, USA), each leaf sample had its NIR spectrum obtained in absorbance mode in an investigated range of wavelengths from 900 to 1700 nm and at a 1.32 nm resolution. To obtain the spectra, we followed a sampling model where the blade of the leaves was removed from the midrib and folded so that the instrument could acquire the spectral information (Fig 1D). In this way, the light emitted by the device would not cross the leaf and cause unwanted dispersion.

Before applying the dissimilarity measurement algorithms, the NIR spectrum matrix was subjected to different pre-treatments [8,22] and some combinations of them to verify the best

**Table 1. Definition of classes referring to pairs of leaf samples used in comparisons according to the plot from which the samples were obtained.**

| Class | Family | Block | Description |
|-------|--------|-------|-------------|
| C1 | The same | The same | The leaves collected represent samples from the same individual |
| C2 | The same | Different | The leaves collected represent samples from different individuals, but from the same family |
| C3 | Different | The same or different | The collected leaves represent samples of individuals from different families and, therefore, different individuals. |

procedure for this type of analysis. Each pre-treatment was tested to verify if the differentiation between the samples would be potentiated after the spectra pre-treatment. In the end, in addition to the original spectrum matrix, the following transformations were tested: Savitzky-Golay Smoothing (SG), First Derivative (D1), Second Derivative (D2), and Multiplicative Scattering Correction (MSC), in addition to the Mean Centering pre-processing (MC) [8]. The pre-treatments combinations included: SG+MC, SG+MSC+MC, SG+D1+MC, SG+D1+MSC+MC, SG+D2+MC, and SG+D2+MSC+MC.

## Statistical framework

The analyses were carried out in two stages to achieve this research's objectives.

Initially, we evaluated the discrimination potential of different individuals, or, conversely, the identification of identical individuals, that is, from the exact clone, based on data from RGB images and NIR spectra. Once the possibility of using images or NIR to classify clones was identified (Table 1), the second stage of the research was to develop a quick and practical protocol for this purpose.

Considering that, at the field level, it would be easier to obtain images of the leaves and guarantee the success of step 1, the second step of this research consisted of the development of a protocol for collecting images of two samples of leaves for decision making regarding whether they belong to the same or different individuals. The protocol described in this work was tested on a subset of the images.

**Euclidean distance (*D*) as a measure of dissimilarity.** The Euclidean distance [19,23] was used to compare the attributes of pairs of images organized into three distinct classes (Table 1). In the first stage of this research, the leaf attributes were obtained in two ways: RGB and NIR. Additionally, these attributes were aggregated to verify the eventual sum of effects and, therefore, the power of sample discrimination.

The discrimination of samples according to the Euclidean Distance of the attributes obtained based on NIR, RGB and NIR+RGB was evaluated under different scenarios (Table 2). ROC (Receiver Operating Characteristic) curves [24], representing the relationship between false positive and true positive rates, were used for the comparison. The determination of the ROC curve and the area under the curve (AUC—a measure of the discriminatory capacity of the classification algorithm) is related to the construction of the confusion matrix and the calculation of sensitivity and specificity measures [25–27]. Models with higher AUC are better in terms of accuracy or, in our case, discriminatory power [24]. It is noteworthy that, in the present study, all attributes were standardized prior to data analysis. Standardization of variables is necessary when they have different dimensions [24]. In the present study, we used image data and NIR spectra. Our standardization was carried out in order to assign mean 0 and standard deviation equal to 1 to all variables.

**Table 2. Scenarios used in classifying samples regarding the corresponding Euclidean distance values obtained between pairs of images or NIR spectra.**

| Scenarios | Origin of attributes used |
|---|---|
| RGB | RGB image |
| R | Red from RGB image |
| NIR | NIR spectrum after pre-treatment |
| RGB+NIR | RGB image combined with pre-treated NIR spectrum |
| R+NIR | Red combined with pre-treated NIR spectrum |

RGB, Red, Green, or Blue obtained from digital images; NIR, near-infrared.

**Mahalanobis distance ($D^2$) as a measure of dissimilarity.** The Mahalanobis distance [19,23] was used in the work's second step to develop a quick and straightforward protocol for decision-making on classifying pairs of RGB images collected in the field. This protocol would be necessary if we have images of stalks under selection in the SS system that are sufficiently close to each other, which could lead to doubts about whether such clones could come from the same plant.

The proposed protocol is to compare two images using the following procedure:

1. Collect the +1 leaf image of two vigorous stalks located so close together that it may seem suspicious whether or not the stalks belong to the same individual;

2. Take an image from each leaf and use a resampling [24] approach to select a number *n* of pixels from each image; n is sufficiently large and smaller than the total number of pixels in the image. Each sampling of pixels will compose a new image, called the pixel-resampling image of the original image;

3. Obtain the average vector of the attributes of each pixel-resampling image;

4. Calculate the differences between the attributes obtained from pairs of pixel-resampling images;

5. Obtain the same attributes from each of the two original images;

6. Obtain the average vector of the attributes of each original image;

7. Calculate the difference between the attributes obtained in the previous step;

8. Construct the combined covariance matrix of the attribute differences from steps 4 and 7;

9. Calculate the Mahalanobis distance between the mean vector obtained in step 7 and the vector of the population mean obtained in step 4, weighted by the matrix obtained in 8; the corresponding p-value is evaluated.

Fig 2 outlines the procedure used to aid decision-making regarding the similarity between the two selected stalks.

The rationale presented in this protocol is based on the theory of decomposition (between and within) of the variability between observations of two treatments (or factors) under consideration, the so-called Analysis of Variance (ANOVA) [28,29]. In the case of ANOVA, the difference between the means of two treatments (in our case, two original images) is evaluated by comparing their variability (variability between) with the variability arising from repetitions of the treatments (variability within) using the F test. In our work, we will use the chi-square statistic with $p$ (where $p$ = number of attributes) degrees of freedom for the evaluation of Mahalanobis statistic $D^2$ [19,23]. The statistic $D^2$ is used to evaluate the difference between the vector of differences for the attributes of the original images ("between") and the mean vector of the differences for the attributes of the numerous pixel-resampling images obtained from each image ("within"), weighted by the combined covariance matrix of the attributes of all images.

To obtain the p-value associated with the $D^2$ statistic, we need to assume that the population under consideration has a multivariate normal distribution so that the values of $D^2 \sim \chi^2(p)$ [23]. We checked visually that the $D^2$ was, in fact, following a chi-squared distribution.

**Computational resources.** The analysis was carried out with the R software [30]. Additional R functions, when necessary, were developed at the Laboratory of Analysis and Research in Applied Statistics (LAPEA, www.lapea.ufv.br).

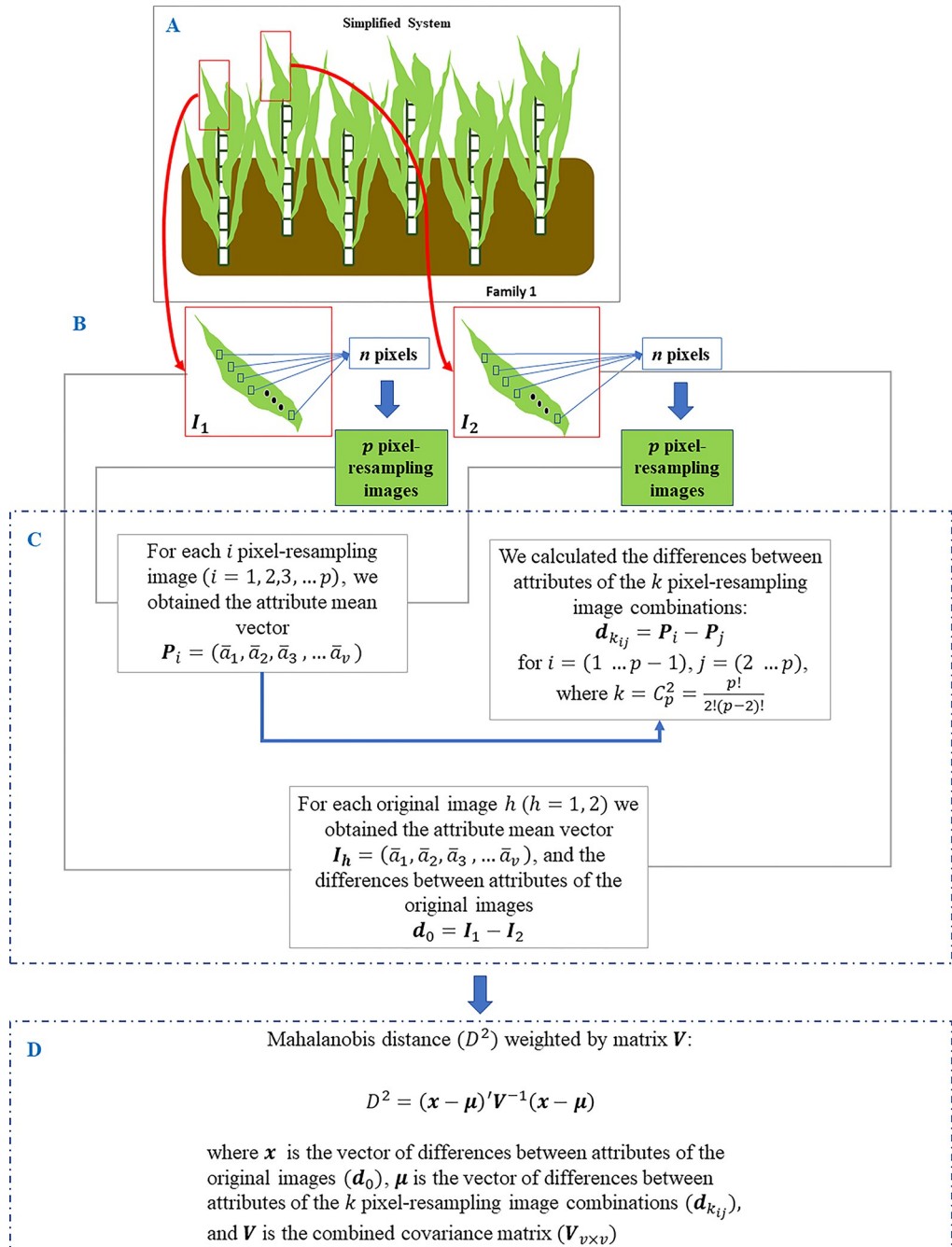

**Fig 2. Protocol used to aid decision-making based on the Mahalanobis distance ($D^2$), and the corresponding p-value.** (A) RGB images are obtained from two suspicious samples. (B) from each image, a large P-number of pixel-resampling images is obtained from the original images. (C) Attributes are taken from the original images and the resulting "fake" images. (D) we obtain the measure $D^2$ and the associated p-value.

## Results and discussion

As commented above, this work had two stages. The first was to assess the possibility of distinguishing between samples from the exact clone or different clones, and the second defined the investigation protocol on pairs of RGB images of suspicious samples.

## Step 1

At this stage of the work, we evaluated using RGB images, NIR spectra, and the combination of both as tools for the discrimination of sugarcane clones.

Initially, we evaluated the use of combined R, G, and B bands (Fig 3A), only R (Fig 3B), and only NIR (Fig 3C) in the discrimination of images previously classified as class C1 (samples from the exact clone) and their complement, that is, all results from classes C2 and C3 (samples from different clones, whether from the same family or not). In this preliminary analysis, it can be seen that there is a concentration of lower values of Euclidean distances when the samples come from the exact clone (as can be seen by observing the median value in each boxplot), as would be expected. Such results are indicators that it would be possible, in practice, to find a way to discriminate clones based on the Euclidean distance of these attributes, as used elsewhere in other applications [14,15,17,31,32]. It is important to emphasize that the change in the axis labels in Fig 3A, 3B and 3C is due to the number of variables involved in calculating the Euclidean distance. In Fig 3A, three variables (R, G, and B) were used; in Fig 3B, only one variable (R); and in Fig 3C, there were 605 variables (corresponding to wavelengths). The different number of variables are causing the changes among the axis labels.

It is essential to inform that the identification of differences between the samples was improved after the application of pre-treatment on the NIR spectra matrix, possibly due to the increase in the signal-to-noise ratio after the application of certain pre-treatments [33,34]. In the current data set, the best results occurred after the Savitzky-Golay Smoothing (with window size 5 and degree 2 polynomial), First Derivative, and Multiplicative Scatter Correction (MSC) transformations, followed by Mean Centering pre-processing method. These pre-treatments are very common in sugarcane works [22,34–36].

To better compare the different alternatives of the sample pairs classification process, we constructed graphs (Fig 4) showing the ROC curves corresponding to the scenarios defined in Table 2.

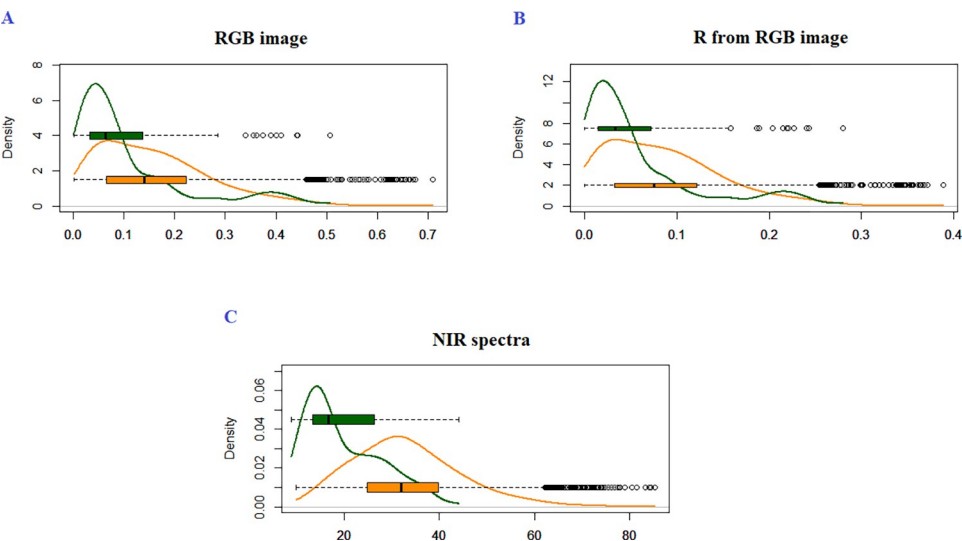

**Fig 3. Distribution of Euclidean distances (*D*) for pairs of C1 (green) and non-C1 (orange) class samples.** (A) Using R, G, and B bands as attributes obtained from leaf images. (B) Using only the R band as an attribute of these images. (C) Using of attributes corresponding to the wavelengths of the pre-treated NIR spectra of leaf samples. The boxplots inserted in the graphs are helpful for better interpreting the descriptive measures, such as the median, of the *D*. C1: The leaves collected represent samples from the same individual within a family.

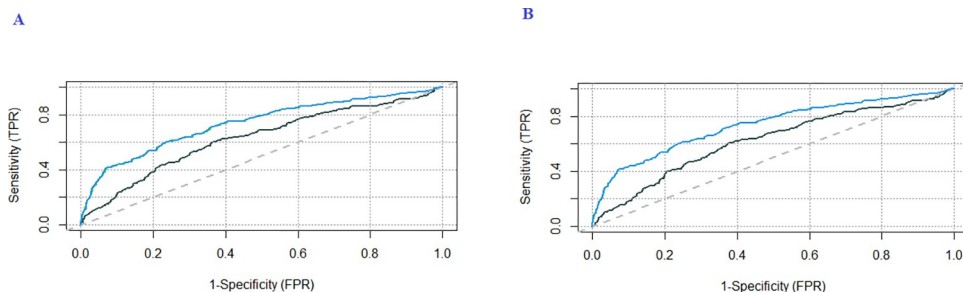

**Fig 4. ROC curves for clone differentiation procedures based on RGB images and near-infrared (NIR) spectra.**
(A) use of attributes R, G, and B (black line), NIR (green), and the combination of RGB and NIR (blue line). (B) use only the attribute R (black line), NIR (green), and the combination of R and NIR (blue line). It is essential to mention that the green and blue lines are superimposed because they present similar results. TPR: True positive rate. FPR: False positive rate.

The ROC curves of the different procedures (Fig 4) indicate their potential for discriminating pairs of samples since such curves are above the 45-degree line corresponding to random classification [24]. It is also observed that using only the R band (AUC = area under the curve = 0.6219) offers practically the same power of discrimination as all image bands (AUC = 0.6289). On the other hand, the greatest discriminatory power would occur using attributes obtained from the NIR (AUC = 0.7348) data. Furthermore, the concatenation of NIR and RGB data (AUC = 0.7360) or NIR and R (AUC = 0.7352) did not significantly improve the clone discrimination process, probably because there are many more attributes from the NIR spectra (605 wavelengths) compared to the attributes derived from the images. Because NIR instruments capture information at the molecular level in leaves [8] and not just color variations in the visible spectrum, better performance was expected from using attributes in NIR.

In all evaluated scenarios (Fig 4), the ROC curves indicate that the best results occurred for the NIR after pre-treatment of the spectrum matrix. However, given the difficulty of having portable instruments in breeding programs for the rapid collection of spectra, in addition to the need to evaluate different pre-treatments in the spectra matrix, which can be a difficulty in the analysis, the use of NIR data turns not to be a good option for use in practice.

Thus, from a practical point of view, the easiest way would be to collect RGB images at the field level, which justifies the development of an image collection and analysis protocol for stalks discrimination in the initial stages of the SS.

## Step 2

At this stage of the work, we used only the RGB images to evaluate the second objective of this research. In particular, considering the previous analysis already discussed in the previous item, it was identified that only the use of the R attribute would give satisfactory results.

$D^2$ values and corresponding p-values were obtained from the analysis of 861 image combinations between pairs of samples from the seven families that presented genotypes in the two experimental blocks under study. Of these 861 pair combinations, there were 42 from class C1, 63 from class C2, and 756 from class C3, as defined in Table 1. It is essential to mention that in our analysis, after testing different numbers of pixel-resampling images (varying from 50 to 1,000), we decided to use 100 pixel-resampling images.

Considering the asymmetry obtained between the $D^2$ values, and between the p-values of the image pairs, we chose to calculate the median of these values in the observed data. For $D^2$, the median of class C1 was the lowest (0.8149) when compared to the medians of classes C2 (2.6163) and C3 (4.7439). As expected, the median p-value for class C1 (0.3744) was higher

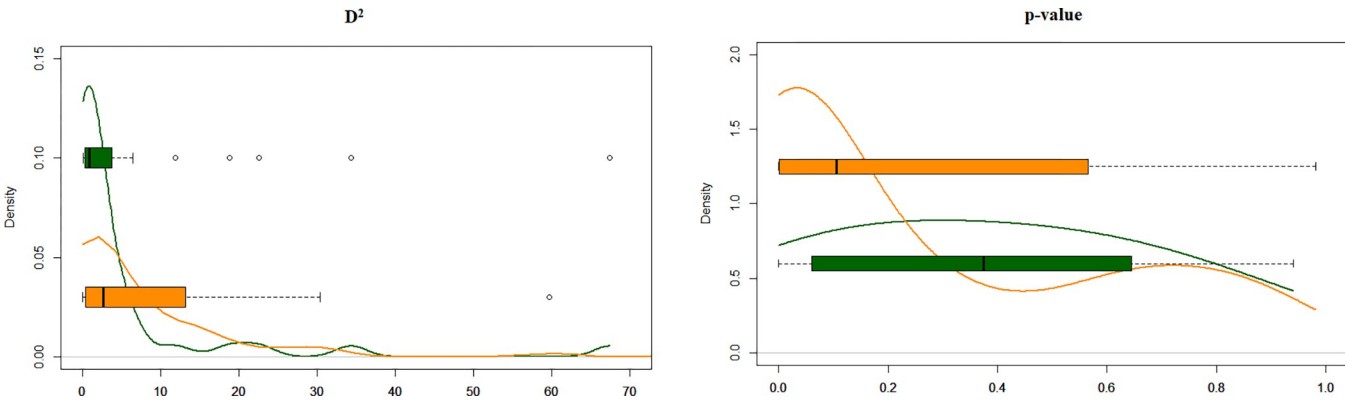

**Fig 5. Distribution of Mahalanobis distance ($D^2$) and p-values for comparing the differences between samples' attributes.** Class C1 (orange): The leaves collected represent samples from the same individual. Class C2 (green): The collected leaves represent samples from individuals from the same family.

than that of the other classes (0.1057 and 0.0294 for classes C2 and C3, respectively). These results confirm the usefulness of the proposed protocol (Fig 2) for genotype discrimination.

On the other hand, comparing individuals from different families is not that important in practice since, in the SS, the families are planted far from each other [5]. The most challenging problem is differentiating clones within the same family in this SS. It can happen that two vigorous individuals within the same family, and which would therefore be selected, are so close to each other that we would be in doubt whether they are shoots of the same plant, that is, a single individual, or whether in fact, they are distinct individuals. To assess this issue, we compared the results for individuals in class C1 against those in class C2 (Fig 5).

Important to mention that the points appearing in Figs 3 and 5 are not necessarily outliers. They are just extreme points that may appear accordingly to the underline distribution of the statistic we depict on the plot. Of course, some of these points could be outliers for some distribution depending on other factors affecting the samples, like disease spots and nutritional aspects of the leaves.

Considering that the value of $D^2$ is affected by the number of attributes in the data matrix, a future investigation would try to identify the p-value threshold that would separate the images of different leaves of the same individual from the rest. If it is impossible to find this threshold, at least the researcher can decide the similarity of the compared stalks. The argument will be that if the images are from the same genotype, collected from different leaves, the distance $D^2$ should be zero. Logically, it would never be zero, as the images would never be identical, but a certain threshold $h$ would make the probability of concluding that two images correspond to the exact clone occur if the distance is smaller than $h$. In our work, we did not try to find a threshold value $h$ for $D^2$, since this value can be influenced by the number of attributes extracted from the images. However, we will consider the p-value associated with the calculated $D^2$ as an additional measure for decision-making. If the p-value associated with the $D^2$ obtained from the comparison of the two images is large (and, therefore, with a relatively small $D^2$), we would indicate that the images are similar and with a high probability of belonging to the same individual. On the other hand, if the p-value is small, there is an indication that the images were obtained from different individuals.

## Conclusions

The Euclidean distance of the attributes derived from RGB images and NIR spectra allows for the discrimination of stalks suspected of being parts of the same individual within the family

being evaluated in the Simplified System. The distinction is made more efficient by using NIR spectra. However, as it is more practical in the field, RGB images would be preferred. The proposed protocol for the inference about any two images proved to be efficient in classifying them as to whether they belong to the same individual and is, therefore, useful in choosing stalks in the Simplified System.

## Acknowledgments

The authors would like to thank the Interuniversity Network for the Development of the Sugarcane Industry (RIDESA) for providing the data and field services carried out in the UFV's sugarcane breeding program, the National Council for Scientific and Technological Development (CNPQ) and FINEP for the research grants, and the Coordenação de Aperfeiçoamento de Pessoal de Nível Superior—Brasil (CAPES) for financing this manuscript.

## Author Contributions

**Conceptualization:** Luiz Alexandre Peternelli.

**Data curation:** Luiz Alexandre Peternelli, Andréa Carla Bastos Andrade.

**Formal analysis:** Luiz Alexandre Peternelli, Andréa Carla Bastos Andrade.

**Funding acquisition:** Luiz Alexandre Peternelli.

**Investigation:** Luiz Alexandre Peternelli, Andréa Carla Bastos Andrade.

**Methodology:** Luiz Alexandre Peternelli, Andréa Carla Bastos Andrade.

**Project administration:** Luiz Alexandre Peternelli.

**Resources:** Luiz Alexandre Peternelli.

**Software:** Luiz Alexandre Peternelli, Andréa Carla Bastos Andrade.

**Supervision:** Luiz Alexandre Peternelli.

**Validation:** Luiz Alexandre Peternelli, Andréa Carla Bastos Andrade.

**Visualization:** Luiz Alexandre Peternelli, Andréa Carla Bastos Andrade.

**Writing – original draft:** Luiz Alexandre Peternelli, Andréa Carla Bastos Andrade.

**Writing – review & editing:** Luiz Alexandre Peternelli, Andréa Carla Bastos Andrade.

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
