## [Decision Letter · Decision Letter 0]

24 Jan 2023

PONE-D-22-33698Insights and protocols for discrimination of sugarcane clones by dissimilarity measures on RGB and NIR dataPLOS ONE

Dear Dr. peternelli,

Thank you for submitting your manuscript to PLOS ONE. After careful consideration, we feel that it has merit but does not fully meet PLOS ONE’s publication criteria as it currently stands. Therefore, we invite you to submit a revised version of the manuscript that addresses the points raised during the review process.

We look forward to receiving your revised manuscript.

Kind regards,

Clara Sousa

Academic Editor

PLOS ONE

Journal Requirements:

2. Please expand the acronym “CNPq” as indicated in your financial disclosure so that it states the name of your funders in full.

Reviewers' comments:

Reviewer's Responses to Questions

**Comments to the Author**

1. Is the manuscript technically sound, and do the data support the conclusions?

Reviewer #1: Yes

Reviewer #2: Partly

Reviewer #3: Partly

Reviewer #4: Partly

2. Has the statistical analysis been performed appropriately and rigorously? 

Reviewer #1: Yes

Reviewer #2: Yes

Reviewer #3: N/A

Reviewer #4: Yes

3. Have the authors made all data underlying the findings in their manuscript fully available?

Reviewer #1: Yes

Reviewer #2: Yes

Reviewer #3: Yes

Reviewer #4: Yes

4. Is the manuscript presented in an intelligible fashion and written in standard English?

Reviewer #1: Yes

Reviewer #2: Yes

Reviewer #3: Yes

Reviewer #4: Yes

5. Review Comments to the Author

Reviewer #1: Original article with an excellent method of contributing to genetic improvement, thinking about large-scale phenotyping strategies. Strategies using RGB images can facilitate the use of these methodologies, when obtained with good calibrations.

Reviewer #2: Manuscript Number: PONE-D-22-33698

Insights and protocols for discrimination of sugarcane clones by dissimilarity measures on RGB and NIR data

The manuscript describes the application of computer vision (RGB images) and NIR spectra to identify sugarcane clones with specific purposes. The work is interesting from the agricultural point of view. There is novelty and the application may provide great support for the task of clone identification. However, the approach proposed should be further detailed, both to provide deeper understanding of the methods, and to allow other researchers to perform the task and apply the method. Hence, I have some suggestions presented below:

Material and methods

It is not clear the number of samples analyzed. It seems that 14 and 24 clones were analyzed, which is a small number from the agricultural point of view. Extracting more pixels could be sufficient from the computer science point of view, but the authors must clarify this information.

The authors should provide more details about the methods, so that is may allow other researchers to perform the experiments or move a step further and not use them if that is the case. Hence, in the next paragraphs I list some details that must be presented:

Regarding the image acquisition procedure, please present further details about the illumination and configuration of the camera (ISO, opening, etc).

Please either add some images in the supplementary material, or report how the region of interest was extracted, as by figure 1 the image contained sample and background. Was it performed manually or automatically using an algorithm?

Lines 115=116: Please mention the pre-treatment methods used and combinations

Line 164: how many resampled images were used?

In line 278, the authors mention 2601 combinations of image resampling. Is this the same number of samples used for NIR statistical analysis? How the authors compared the number of samples for each method? What about when the methods were combined?

Results and discussion

Perhaps the authors could present the confusion matrix for the samples classified, in addition to f-score, recall, as it may provide further support for data interpretation by readers, in addition to the ROC curve.

The spectra of samples should be presented, and the main wavelengths influenced by the samples, related to the organic bonds, should be discussed. This could provide a deeper interpretation of the influence of samples in the NIR spectra, and also further support for future research.

I hereby suggest a few works that may aid the authors in this discussion as they used both image analysis and NIR spectroscopy for similar tasks, but the authors may feel free to find others if suitable:

Computer vision system and near-infrared spectroscopy for identification and classification of chicken with wooden breast, and physicochemical and technological characterization - https://doi.org/10.1016/j.infrared.2018.11.036

Assessment of beer quality based on foamability and chemical composition using computer vision algorithms, near infrared spectroscopy and machine learning algorithms -

https://doi.org/10.1002/jsfa.8506

Automated grapevine cultivar classification based on machine learning using leaf morpho-colorimetry, fractal dimension and near-infrared spectroscopy parameters - https://doi.org/10.1016/j.compag.2018.06.035

Similarly, please discuss how the channel red may have contributed to identifying the desired samples. Please present some grayscale images in this channel for readers information, and how the wavelength in the visible range may identify the samples.

Minor comments

The text should be fully revised for academic writing. Several sentences must be corrected, along with some typos. I hereby list a few of them, although there are others:

I suggest the authors to verify in the agricultural field, either with some researcher or other papers, the proper term for sugarcane stalk, as I believe that ‘stem’ is more commonly used.

Line 20: I believe RGB stands for red, green blue. Please check ‘network’

Line 36: I believe that ‘task’ is more appropriate than ‘job’ in this context

Line 111: ‘the instrument could be read’ or the instrument could acquire the spectral information’?

Line 112: instead of energy fired, please use the light emitted

Line 268: what does ‘passes’ mean? Please rephrase the sentence to clarify the information

Reviewer #3: The manuscript “Insights and protocols for discrimination of sugarcane clones by dissimilarity measures on RGB and NIR data” has several scientific findings:

In all the evaluated scenarios R, G, and B (black line), NIR (green), and the combination of RGB and NIR (blue line), the ROC curves indicate that the best results occurred for the NIR after pre-treatment of the spectrum matrix.

However, the Authors have concluded that with the difficulty of having portable instruments in breeding programs for the rapid collection of spectra, in addition to the need to evaluate different pre-treatments in the spectra matrix, which can be a difficult in the analysis, the use of NIR data passes not to be a good option for use in practice.

This is an abrupt conclusion without any supporting evidence. Although, classifiers are able to distinguish all classes, the AUC value for RGB images (R band (AUC = 0.6219) RGB together (AUC = 0.6289) are lower than NIR (AUC = 0.7348) and NIR and RGB data (AUC = 0.7360). Hence, a classifier with NIR and RGB data would be better to utilize.

Only the use of the R attribute would give more satisfactory results than RGB together.

The analysis of 2601 image combinations between pairs of samples from seven families classified 36 images from class C1, 63 from class C2, and 2502 from class C3 (represent samples of individuals from different families and, therefore, different individuals based on〖 D〗^2 values and corresponding p-values. Which can be differentiated by the traditional ways too.

Although, the objective of the present study is to differentiate clones (C2- represent samples from different individuals, but from the same family) within the same family in this SS, the present study suggests that this class C2 has very low representations in terms of the number of images in total of 2601 images.

The legend of Fig. 3 is not clear, please elaborate it.

Please explain about outliers in each fig 3 and 4.

Overall, a good attempt was made to distinguish the individuals from the same and different families, however, the work is still in the preliminary stage and needs extensive study with more data before publication.

Reviewer #4: The proposed manuscript addresses the use of RGB and NIR data to discriminate between sugarcane clones and families by the use of Euclidean and Mahalanobis distance dissimilarity measures. It is written in good English and presents an adequate structure. It seems technically sound for the most part, with a reasonable description of the employed methodology (though it can be improved), and withdraws conclusions only partly supported by the presented data. Hence, my recommendation is for a Major revision.

A reviewer attachment file is provided with the suggested corrections and modifications to the proposed manuscript. Mainly the authors should focus on: need and method for standardization of the employed metrics; normal distribution assumption correctness; in depth method description and results; in depth analysis of the obtained p-values.

6. PLOS authors have the option to publish the peer review history of their article (what does this mean?). If published, this will include your full peer review and any attached files.

Reviewer #1: No

Reviewer #2: No

Reviewer #3: **Yes: **Rasappa Viswanathan

Reviewer #4: No

---

## [Author Response · Author response to Decision Letter 0]

10 Mar 2023

Rebuttal letter to the academic editor and reviewers (There is an attached file containing the same text as presented here)

We want to thank all for the suggestions, comments, and criticisms regarding our manuscript. Your contributions will, for sure, improve the manuscript. Below you can find, in red, the authors’ responses for each question or comment. The highlighted changes were made to the ‘Revised Manuscript with Track Changes’ file.

Journal Requirements:

2. Please expand the acronym “CNPq” as indicated in your financial disclosure so that it states the name of your funders in full.

Review Comments to the Author:

Reviewer #1: 

i. Original article with an excellent method of contributing to genetic improvement, thinking about large-scale phenotyping strategies. Strategies using RGB images can facilitate the use of these methodologies, when obtained with good calibrations.

Authors: Thank you for your consideration regarding the potential contribution of our manuscript.

Reviewer #2: 

Material and methods

ii. It is not clear the number of samples analyzed. It seems that 14 and 24 clones were analyzed, which is a small number from the agricultural point of view. Extracting more pixels could be sufficient from the computer science point of view, but the authors must clarify this information.

Authors: Considering the sample as a clone, 38 samples were analyzed. Each sample contained images of three sheets, totaling 114 images. We believe that these 114 images would be sufficient for this work, that is, to discriminate individuals based on dissimilarity measures and that meet the classes defined in Table 1. Considering that the algorithm worked for a smaller sample size, we concluded it was efficient for our purpose. 

iii. The authors should provide more details about the methods, so that is may allow other researchers to perform the experiments or move a step further and not use them if that is the case. Hence, in the next paragraphs I list some details that must be presented:

a. Regarding the image acquisition procedure, please present further details about the illumination and configuration of the camera (ISO, opening, etc).

Authors: In fact, it is essential to have as much information as possible so that any other researcher can reproduce the work. In our case, we ensured that the lighting was the same for obtaining the images, as specified in line 106 of the Material and Methods (original manuscript). Since the camera-to-object distance conditions were constant, we used the digital camera's automatic option. We included this information in the text (lines 109 and 110).

b. Please either add some images in the supplementary Material, or report how the region of interest was extracted, as by figure 1 the image contained sample and background. Was it performed manually or automatically using an algorithm?

Authors: We can report how the segmentation was performed to obtain the interest region of the image. We used segmentation from package ‘pliman’. We included this information in the text (lines 110 and 111).

c. Lines 115=116: Please mention the pre-treatment methods used and combinations.

Authors: The pre-treatments used in the rows of the spectrum matrix were informed, with only the combinations remaining to be informed (lines 119 to 121, original manuscript). Considering that in this type of work involving NIR, not all the combinations used are specified, we present only the one that gave the best result for our dataset. This information appears in lines 236 to 239 of the results and discussion (original manuscript). As a way of helping the researcher to identify the pre-treatment combinations we used, we added this information in the Material and Methods (lines 126 to 128). 

d. Line 164: how many resampled images were used?

Authors: On line 164, the number of resampled images is not specified (what we call fake images) since, at this point, we are presenting only the algorithm. In our case, after testing different numbers of fake images (varying from 50 to 1,000), we decided to use 100 fake images. We apologize for not including this information in the results and discussion section. We include this information now in the section Results and Discussion (Step 2), in lines 295 to 297. Notice that we replaced the expression “fake images” with “pixel-resampling images,” as suggested by another reviewer.

e. In line 278, the authors mention 2601 combinations of image resampling. Is this the same number of samples used for NIR statistical analysis? How the authors compared the number of samples for each method? What about when the methods were combined?

Authors: There are not 2601 combinations, but 861 combinations corresponding to the samples of the seven families that presented genotypes in two experimental blocks. After rereading this section, we noticed and corrected this error in the manuscript. Answering your question, if the samples match the same number of samples used for NIR, the answer is no. We evaluated using NIR in the first part of the manuscript (Step 1). In this case, we considered more samples, as presented in the item 'Plant material' of Material and Methods (lines 81 to 84 of the original manuscript). In Step 1, 114 samples (38 clones x 3 leaves sampled per clone) were collected. Therefore, 6441 pairs of samples were tested. In the case of Step 2 of the objectives, we selected only those families with representatives in both blocks, as mentioned earlier. They totaled 861 combinations (14 clones x 3 sampled leaves, taken two by two).

Results and discussion

i. Perhaps the authors could present the confusion matrix for the samples classified, in addition to f-score, recall, as it may provide further support for data interpretation by readers, in addition to the ROC curve.

Authors: There is no way to create a single confusion matrix, as 6,441 pairs of samples were evaluated. Furthermore, there was no optimal Euclidean Distance threshold for discriminating between samples from different individuals. Therefore, we used ROC curves to assess the discriminatory power of using images, NIR, or a combination of both.

ii. The spectra of samples should be presented, and the main wavelengths influenced by the samples, related to the organic bonds, should be discussed. This could provide a deeper interpretation of the influence of samples in the NIR spectra, and also further support for future research.

Authors: If the objective were to use the NIR in predicting some attribute of agronomic interest (for example, fiber content, sucrose content), the association between wavelengths and organic bonds would be interesting. Our research group is developing future studies involving this type of investigation. However, for the present manuscript, our main objective was to evaluate sample discrimination between individuals only. Therefore, we did not state any discussion regarding that matter. 

iii. I hereby suggest a few works that may aid the authors in this discussion as they used both image analysis and NIR spectroscopy for similar tasks, but the authors may feel free to find others if suitable: 

a. Computer vision system and near-infrared spectroscopy for identification and classification of chicken with wooden breast, and physicochemical and technological characterization - https://doi.org/10.1016/j.infrared.2018.11.036

b. Assessment of beer quality based on foamability and chemical composition using computer vision algorithms, near infrared spectroscopy and machine learning algorithms - https://doi.org/10.1002/jsfa.8506

c. Automated grapevine cultivar classification based on machine learning using leaf morpho-colorimetry, fractal dimension and near-infrared spectroscopy parameters - https://doi.org/10.1016/j.compag.2018.06.035

Authors: We appreciate the reading suggestions. They present relevant contributions to research with NIR and in different applications. In general, the studies bring insights into prediction and classification through the interpretation of wavelengths concerning the quality of food and animal and plant origin products, as well as genotypic differentiation of agronomic cultivars. However, as previously mentioned, the primary aim of our research is only to differentiate genotypes and not to associate them with a specific trait. This topic is another front of study under development by our research group. The suggested papers will undoubtedly contribute to future ideas and discussions.

iv. Similarly, please discuss how the channel red may have contributed to identifying the desired samples. Please present some grayscale images in this channel for readers information, and how the wavelength in the visible range may identify the samples.

Authors: The entire selection process was done independently and through trial and error. When testing R alone, we found that the results were better than combining the three colors or using G and B alone. This result somehow agrees with the literature, which mentions that the blue color contributes little to the vegetation study. For this reason, some camera manufacturers give the options of changing band B to NIR on their devices (see for example: https://www.mapir.camera/en-gb/pages/survey3-cameras#specs).

Minor comments

The text should be fully revised for academic writing. Several sentences must be corrected, along with some typos. I hereby list a few of them, although there are others:

I suggest the authors to verify in the agricultural field, either with some researcher or other papers, the proper term for sugarcane stalk, as I believe that ‘stem’ is more commonly used.

Line 20: I believe RGB stands for red, green blue. Please check ‘network’

Line 36: I believe that ‘task’ is more appropriate than ‘job’ in this context

Line 111: ‘the instrument could be read’ or the instrument could acquire the spectral information’?

Line 112: instead of energy fired, please use the light emitted

Line 268: what does ‘passes’ mean? Please rephrase the sentence to clarify the information

Authors: Thanks for the suggestions and pinpoint corrections. We made these corrections to the text, where they appear. We also rerun a new writing revision to the whole manuscript. For the first suggestion, regarding using the term 'stalk' versus 'stem,' we prefer the term 'stalk' as they appear in the majority of the literature on sugarcane.

Reviewer #3: 

i. The manuscript “Insights and protocols for discrimination of sugarcane clones by dissimilarity measures on RGB and NIR data” has several scientific findings:

a. In all the evaluated scenarios R, G, and B (black line), NIR (green), and the combination of RGB and NIR (blue line), the ROC curves indicate that the best results occurred for the NIR after pre-treatment of the spectrum matrix.

However, the Authors have concluded that with the difficulty of having portable instruments in breeding programs for the rapid collection of spectra, in addition to the need to evaluate different pre-treatments in the spectra matrix, which can be a difficult in the analysis, the use of NIR data passes not to be a good option for use in practice.

This is an abrupt conclusion without any supporting evidence. Although, classifiers are able to distinguish all classes, the AUC value for RGB images (R band (AUC = 0.6219) RGB together (AUC = 0.6289) are lower than NIR (AUC = 0.7348) and NIR and RGB data (AUC = 0.7360). Hence, a classifier with NIR and RGB data would be better to utilize.

Only the use of the R attribute would give more satisfactory results than RGB together. 

Authors: We appreciate the considerations presented concerning our scientific findings. Our research had two distinct objectives. The results indicated the possibility of discrimination of individuals at the field level and that the NIR+RGB combination would be better, as informed in the conclusion of the work (objective 1). For practicality and our own experience, we consider the approach using only RGB images more appropriate for any breeding program (objective 2). The reason is that it would not require a specific instrument for collecting NIR spectra in the field, and also a need to have an expert running the NIR analysis and interpretation, which is much more challenging to be performed.

ii. The analysis of 2601 image combinations between pairs of samples from seven families classified 36 images from class C1, 63 from class C2, and 2502 from class C3 (represent samples of individuals from different families and, therefore, different individuals based on〖 D〗^2 values and corresponding p-values. Which can be differentiated by the traditional ways too.

Although, the objective of the present study is to differentiate clones (C2- represent samples from different individuals, but from the same family) within the same family in this SS, the present study suggests that this class C2 has very low representations in terms of the number of images in total of 2601 images. 

Authors: We used only the seven families with representatives in two experimental blocks to verify our genotype discrimination procedure. We needed samples meeting the three classes of genotypes, as shown in Table 1, just to be able to check for all these possibilities. However, we did consider only the individuals from classes C1 and C2 since they represent the field's most challenging situation. This information appears on lines 304 to 310 within the Results and Discussion section. 

iii. The legend of Fig. 3 is not clear, please elaborate it. 

Authors: We have included more information on the legend of Fig. 3.

iv. Please explain about outliers in each fig 3 and 5. 

Authors: the points appearing on the plot do not necessarily are outliers. They are just extreme points that may appear accordingly to the underline distribution of the statistic we depict on the plot. Important to mention that the orange boxplot in Fig. 3 is a mixture of C2 and C3 classes. Of course, some of these points could be outliers for some distribution depending on other factors affecting the samples, like disease spots and nutritional aspects of the leaves. We have included this discussion in the text (lines 311 to 315).

Reviewer #4: 

A reviewer attachment file is provided with the suggested corrections and modifications to the proposed manuscript. Mainly the authors should focus on: need and method for standardization of the employed metrics; normal distribution assumption correctness; in depth method description and results; in depth analysis of the obtained p-values.

i. Line 103: Doesn’t leaf coloration change with culture time? Were the samples collected at the same culture day? 

Authors: In fact, leaf coloration may change with crop time. The leaves should be vigorous, healthy, and green. We lack this information in the manuscript. To better explain this detail, we included more information on lines 104 and 105, explaining that the samples were collected in the same week.

ii. Line 146 (area under the curve): Please further explain how this parameter was used in this work. 

Authors: Thanks for pointing out this aspect of the AUC parameter. We have included a further explanation on lines 156 and 157.

iii. Line 149 (all attributes were standardized): Please elaborate more on the need for standardization for the employed metrics (aren’t some of them already standardized by default?) and the method employed. 

Authors: We included further explanation regarding the standardization used in our paper and a literature where the reader could achieve extra information. This information appears in lines 158 and 161.

iv. Line 168 (fake): “Fake image” seems odd to me, what about “resample image”? Authors: “Fake” image was the first idea that came to us during the manuscript writings. However, we agree that this term is odd. Also, since the image was not “resampled,” but the pixels were resampled, we suggest using the term “pixel-resampling image” instead. We hope this will attend your concerns.

v. Line 201 (normal distribution): have the authors determined (by an appropriate statistical method) if this assumption is correct? 

Authors: This statement appears in Mainly (2004). We included this reference on line 215 to support our statement better. However, we did not test this assumption. Instead, we checked visually that the D2 was, in fact, following a chi-squared distribution. This information was included in the manuscript (lines 215 and 216) to make it clear to the reader.

vi. Line 217 (R, G, and B): Were the three images (R, G, and B) analyzed individually or was a combination of the 3 used in a single image used? 

Authors: Initially, in step 1, as stated on line 217 (original manuscript), we analyzed the RGB image, or else, all three bands combined. 

vii. Line 218 (NIR): Please elaborate more. What wavelengths (all, solely a few selected wavelengths, etc.)?

Authors: In this case, we considered the whole spectra, or else, from 900 to 1700 nm, as presented by the instrument. We included extra information about the NIR instrument and the wavelengths: we used a DLP® NIRscan Nano™ EVM spectrometer (Texas Instruments Inc., Dallas, Texas, USA), with a spectral range of 900-1700 nm and at a 1.32 nm resolution. We presented this information on lines 112 to 115 of the revised manuscript.

viii. Lines 255 and 256 (all bands together (AUC = 0.6289)): Doesn’t the blue line in graph 4A represents “all bands (NIR+RGB) together”? And if so, how can the AUC of that line be so close to the AUC of the R band (black line on 4B)? They seen quite apart analyzing figure 4A and 4B.

Authors: Thanks for observing our mistake in not informing that the term “all bands” should be, in fact, “all image bands”, or else, the RGB together. It does not involve NIR in this comparison yet. We have corrected this information in the text (lines 269 and 270).

ix. Line 286 (0.1057 and 0.0579 for classes C2 and C3, respectively): Both these values are higher than 0.05. Doesn’t that the majority of the attributes from individuals from the same family (C2) and from different families (C3) are not statistically different? 

Authors: It would not be statistically different if we fixed the p-values to 0.05 when stating any conclusion. On the other hand, if we use the p-values to measure the casualty of the results, we prefer to interpret it as a measure of the strength of the comparison towards the null hypothesis of no difference between the two samples. In other words, the higher the p-value, the less different the samples are; the smaller the p-value, the more different the samples are. Important to highlight that we slightly corrected the information presented on lines 298 to 304 observed by the other reviewer. Because of this correction, the p-value of class C3 has decreased from 0.0579 to 0.0294, which makes our discussion in the manuscript even better.

x. Line 302 (identify the p-value threshold that would separate): shouldn’t that be 0.05 (for 95% significance)? 

Authors: Not necessarily. In many research areas, like the Medical one, researchers might consider using a minimal value for considering a new drug as better than the actual drug. In the agricultural area, researchers could state a larger p-value (5 to 10%) as necessary information for rejecting a null hypothesis in favor of a new fertilizer or variety, for example. Since we are proposing a new protocol to discriminate samples using only one image of each, from lines 301 to 315 of the original manuscript, we briefly discussed this issue. In this way, we believe the reader could better perceive his findings or even start new research on this subject.

---

## [Decision Letter · Decision Letter 1]

4 Apr 2023

PONE-D-22-33698R1Insights and protocols for discrimination of sugarcane clones by dissimilarity measures on RGB and NIR dataPLOS ONE

Dear Dr. peternelli,

Thank you for submitting your manuscript to PLOS ONE. After careful consideration, we feel that it has merit but does not fully meet PLOS ONE’s publication criteria as it currently stands. Therefore, we invite you to submit a revised version of the manuscript that addresses the points raised during the review process.

We look forward to receiving your revised manuscript.

Kind regards,

Clara Sousa

Academic Editor

PLOS ONE

Journal Requirements:

Reviewers' comments:

Reviewer's Responses to Questions

**Comments to the Author**

1. If the authors have adequately addressed your comments raised in a previous round of review and you feel that this manuscript is now acceptable for publication, you may indicate that here to bypass the “Comments to the Author” section, enter your conflict of interest statement in the “Confidential to Editor” section, and submit your "Accept" recommendation.

Reviewer #2: All comments have been addressed

Reviewer #3: (No Response)

Reviewer #4: All comments have been addressed

2. Is the manuscript technically sound, and do the data support the conclusions?

Reviewer #2: Yes

Reviewer #3: Yes

Reviewer #4: Yes

3. Has the statistical analysis been performed appropriately and rigorously? 

Reviewer #2: Yes

Reviewer #3: Yes

Reviewer #4: Yes

4. Have the authors made all data underlying the findings in their manuscript fully available?

Reviewer #2: Yes

Reviewer #3: Yes

Reviewer #4: Yes

5. Is the manuscript presented in an intelligible fashion and written in standard English?

Reviewer #2: Yes

Reviewer #3: Yes

Reviewer #4: Yes

6. Review Comments to the Author

Reviewer #2: The authors have addressed all comments raised by the reviewers. There is only one issue that I would suggest to be avoided in the future, that is commit to activities in the future and not in the current work, such as reported here:

(Authors: If the objective were to use the NIR in predicting some attribute of

agronomic interest (for example, fiber content, sucrose content), the association

between wavelengths and organic bonds would be interesting. Our research group

is developing future studies involving this type of investigation. However, for the

present manuscript, our main objective was to evaluate sample discrimination

between individuals only. Therefore, we did not state any discussion regarding

that matter.) and here (However, as

6

previously mentioned, the primary aim of our research is only to differentiate

genotypes and not to associate them with a specific trait. This topic is another front

of study under development by our research group. The suggested papers will

undoubtedly contribute to future ideas and discussions.)

I do not see the point of why not doing it in the current study, making a strong research and report it. Instead, it seems more that there will be further similar studies to the current one, this splitting the work and its relevance.

Reviewer #3: Insights and protocols for discrimination of sugarcane clones by dissimilarity measures on RGB and NIR datafocus on early genetic improvement/selection while considering extensive phenotyping techniques. Perhaps, with this data, RGB/or R images can facilitate the selection. I still have few queries:

1. Fig 3 explains clearly about discrimination of clones based on the Euclidean distance. Exact clones C1 showed lower values of Euclidean distances concentration compared to other. Please explain in detail regarding the change in axis labels in 3A, 3B and 3C.

2. Fig 4 suggests that superimposition of NIR (green), and the combination of RGB and NIR (blue line) and author explained that more attributes from NIR spectra (605 wavelengths) leads to overlapping. In such case, can we have a curve of RGB and R together to clear understanding?

3. It is agreed that D2would never be zero and h value should be calculated if researchers want to differentiate ap per D2.Since it has not been done and opined that D2 might be influenced by number of attributes extracted from the images, therefore p-value associated with the D2was referred as an additional measure for decision-making. Although associated with D2 (no significant difference among orange and green line), why doesn't the quantity of attributes affect the p value (significant difference among orange and green line)?

4. Please provide citation which also suggests that p value is better than D2 value.

5. I would like suggest to choose Kullback-Leibler discrepancy as dissimilarity measure.

Reviewer #4: I believe that the revised manuscript may now be considered for publication, though the normal distribution issue is, still, not fully addressed.

7. PLOS authors have the option to publish the peer review history of their article (what does this mean?). If published, this will include your full peer review and any attached files.

Reviewer #2: No

Reviewer #3: **Yes: **Rasappa Viswanathan

Reviewer #4: No

---

## [Author Response · Author response to Decision Letter 1]

17 May 2023

We want to thank all for the suggestions, comments, and criticisms regarding our manuscript. The reviewers' contributions will, for sure, improve the manuscript. Below, one can find, in red, the authors’ responses to each question or comment. The highlighted changes were made to the ‘Revised Manuscript with Track Changes’ file.

Journal Requirements:

Please review your reference list to ensure that it is complete and correct. If you have cited papers that have been retracted, please include the rationale for doing so in the manuscript text, or remove these references and replace them with relevant current references. Any changes to the reference list should be mentioned in the rebuttal letter accompanying your revised manuscript. If you need to cite a retracted article, indicate the article’s retracted status in the References list and also include a citation and full reference for the retraction notice.

Authors: We have reviewed the list of references and have not detected any modifications that should be mentioned.

Review Comments to the Author:

Reviewer #2: 

The authors have addressed all comments raised by the reviewers. There is only one issue that I would suggest to be avoided in the future, that is commit to activities in the future and not in the current work, such as reported here:

 (Authors: If the objective were to use the NIR in predicting some attribute of agronomic interest (for example, fiber content, sucrose content), the association between wavelengths and organic bonds would be interesting. Our research group is developing future studies involving this type of investigation. However, for the present manuscript, our main objective was to evaluate sample discrimination between individuals only. Therefore, we did not state any discussion regarding that matter.) 

and here 

 (However, as previously mentioned, the primary aim of our research is only to differentiate genotypes and not to associate them with a specific trait. This topic is another front of study under development by our research group. The suggested papers will undoubtedly contribute to future ideas and discussions.)

I do not see the point of why not doing it in the current study, making a strong research and report it. Instead, it seems more that there will be further similar studies to the current one, this splitting the work and its relevance.

Authors: We apologize for referring to future work in our response. The present dataset will not be used to infer the association between wavelengths and organic bonds. As we stated before, for the present manuscript, our main objective was to evaluate sample discrimination between individuals only, with no further discussion or inference regarding the inner part of the samples. What we mentioned about future work is still being planned and will be developed by other team members in new experiments planned for this specific purpose. 

Reviewer #3: 

Insights and protocols for discrimination of sugarcane clones by dissimilarity measures on RGB and NIR datafocus on early genetic improvement/selection while considering extensive phenotyping techniques. Perhaps, with this data, RGB/or R images can facilitate the selection. I still have few queries:

 Fig 3 explains clearly about discrimination of clones based on the Euclidean distance. Exact clones C1 showed lower values of Euclidean distances concentration compared to other. Please explain in detail regarding the change in axis labels in 3A, 3B and 3C. 

Authors: In Fig. 3, the change in the axis labels in Figs 3A, 3B, and 3C is due to the number of variables involved in calculating the Euclidean distance. In Fig 3A, three variables (R, G, and B) were used; in Fig 3B, only one variable (R); and in Fig 3C, there were 605 variables (corresponding to wavelengths). The different number of variables are causing the changes between the axis labels. We included this information in the text (lines 236 to 240).

 Fig 4 suggests that superimposition of NIR (green), and the combination of RGB and NIR (blue line) and author explained that more attributes from NIR spectra (605 wavelengths) leads to overlapping. In such case, can we have a curve of RGB and R together to clear understanding? 

Authors: Both curves appear in the graphs: the RGB curve in Fig 4A and the R curve in Fig 4B. As the axes are on the same scale, the comparison is immediate, and it can be seen that they are practically identical.

 It is agreed that D2 would never be zero and h value should be calculated if researchers want to differentiate ap per D2.Since it has not been done and opined that D2 might be influenced by number of attributes extracted from the images, therefore p-value associated with the D2was referred as an additional measure for decision-making. Although associated with D2 (no significant difference among orange and green line), why doesn't the quantity of attributes affect the p value (significant difference among orange and green line)?

Authors: Assuming the same number of attributes, D2 would be considerable if we compared different samples. However, if the samples are very similar, the D2 would be small. The magnitude of D2 is conditioned to the number of attributes used in calculating the statistic. In turn, the p-value that will be associated with the D2 obtained in each case would be obtained from a specific χ^2 distribution, since the distribution is defined by its degree of freedom, which depends on the number of variables. If you are referring to Fig 5, we are visually comparing classes C1 and C2. In both graphs, it is verified that both the D2 and the p-value indicate that individuals from the C1 class tend to give lower results for D2 and higher for p-value, which is consistent with our discussion presented in the article.

 Please provide citation which also suggests that p value is better than D2 value. 

Authors: We are not saying that the p-value will replace the D2 or is better than the D2, but that they can be used together to aid the researcher in decision-making (see comments that follow lines 329 of the original manuscript). This procedure appears in books dealing with statistical inference.

 I would like suggest to choose Kullback-Leibler discrepancy as dissimilarity measure. 

Authors: Thank you for the suggestion. However, as we are working with the idea of comparing samples two-by-two and not comparing probabilistic distributions, according to the suggested application for the use of KL-divergence in Joyce (2011, https://doi.org/10.1007/978-3-642-04898-2_327), we opted for the measures used in the article.

Reviewer #4: 

I believe that the revised manuscript may now be considered for publication, though the normal distribution issue is, still, not fully addressed.

Authors: Thank you for your consideration.

---

## [Editor Report · Decision Letter 2]

18 May 2023

PONE-D-22-33698R2Insights and protocols for discrimination of sugarcane clones by dissimilarity measures on RGB and NIR dataPLOS ONE

Dear Dr. peternelli,

Thank you for submitting your manuscript to PLOS ONE. After careful consideration, we feel that it has merit but does not fully meet PLOS ONE’s publication criteria as it currently stands. Therefore, we invite you to submit a revised version of the manuscript that addresses the points raised during the review process.

We look forward to receiving your revised manuscript.

Kind regards,

Clara Sousa

Academic Editor

PLOS ONE
---

## [Author Response · Author response to Decision Letter 2]

19 Jun 2023

There are no specific questions from reviewers or editor.

---

## [Editor Report · Decision Letter 3]

28 Jun 2023

Insights and protocols for discrimination of sugarcane clones by dissimilarity measures on RGB and NIR data

PONE-D-22-33698R3

Dear Dr. peternelli,

We’re pleased to inform you that your manuscript has been judged scientifically suitable for publication and will be formally accepted for publication once it meets all outstanding technical requirements.

Kind regards,

Clara Sousa

Academic Editor

PLOS ONE

---

## [Editor Report · Acceptance letter]

12 Jul 2023

PONE-D-22-33698R3 

Insights and protocols for discrimination of sugarcane clones by dissimilarity measures on RGB and NIR data 

Dear Dr. peternelli:

I'm pleased to inform you that your manuscript has been deemed suitable for publication in PLOS ONE. Congratulations! Your manuscript is now with our production department. 

Kind regards, 

on behalf of

Dr. Clara Sousa 

Academic Editor

PLOS ONE